# Comparative Personality Traits Assessment of Three Species of Communally Housed Captive Penguins

**DOI:** 10.3390/ani9060376

**Published:** 2019-06-20

**Authors:** Giovanni Quintavalle Pastorino, Richard Preziosi, Massimo Faustini, Giulio Curone, Mariangela Albertini, Dawn Nicoll, Lorna Moffat, Romain Pizzi, Silvia Mazzola

**Affiliations:** 1Department of Veterinary Medicine, Università degli Studi di Milano, Via Celoria 10, 20133 Milan, Italy; giovanni.quintavalle@unimi.it (G.Q.P.); massimo.faustini@unimi.it (M.F.); giulio.curone@unimi.it (G.C.); mariangela.albertini@unimi.it (M.A.); 2Division of Biology and Conservation Ecology, School of Science and the Environment, Faculty of Science and Engineering, Manchester Metropolitan University, Manchester M1 5GD, UK; R.Preziosi@mmu.ac.uk; 3Royal Zoological Society of Scotland, Edinburgh Zoo, 134 Corstorphine Road, Edinburgh EH12 6TS, UK; DNicoll@rzss.org.uk (D.N.); LMoffat@rzss.org.uk (L.M.); romainpizzi@yahoo.com (R.P.)

**Keywords:** *Pygoscelis papua*, *Aptenodytes patagonicus*, *Eudyptes moseleyi*, personality traits

## Abstract

**Simple Summary:**

The evolution of modern zoo structures frequently includes enclosures in which different species are hosted, living together permanently. This choice involves positive aspects, both for visitors and for hosted animals. On the other hand, the space available to animals and the standardization of management procedures may not be sufficiently adequate to allow the behavioral individuality to be maintained. This study aimed to verify, in a colony of three species of penguins living together; if individuals express personality traits, that differ among species. The results we obtained show that the penguins exhibited common personality traits, but also that some were expressed with a different intensity, depending on the specie. From a practical point of view, these data could help the management of the animals, allowing to optimize the design of enclosures and the enrichment strategy according to the different behavioral characteristics of the cohabiting species, in order to match with the needs of the individual.

**Abstract:**

Understanding animal personalities has notable implications in the ecology and evolution of animal behavior, but personality studies can also be useful in optimizing animal management, with the aim of improving health and well-being, and optimizing reproductive success, a fundamental factor in the species threatened with extinction. Modern zoos are increasingly being structured with enclosures that host different species, which permanently share spaces. This condition has undeniable positive aspects, but, in some species, it could determine the appearance of collective or synchronized behaviors. The aim of this study was to verify, in a colony of three species of communally housed penguins (*Pygoscelis papua*, *Aptenodytes patagonicus* and *Eudyptes moseleyi*), through a trait-rating assessment, if interspecific group life impacts on the expression of personality traits, and if it is possible to highlight specie-specific expression of personality traits, despite the influence of forced cohabitation. For many of the personality traits we analyzed, we have observed that it was possible to detect an expression that differed, according to the species. From a practical point of view, these data could ameliorate the management of the animals, allowing to design animal life routines, according to the different behavioral characteristics of the cohabiting species.

## 1. Introduction

Animal personality, defined as behavioral differences between individuals that are consistent over time and across contexts [1], is one of the most emerging subjects in behavioral research [2]. Many authors agree to consider differences in animal personality as the result of adaptive evolutionary processes [3] and to occur in a wide range of taxa [4,5]. Moreover, recent findings suggest that individuals differ consistently in their behavioral tendencies and that the behavior in one context is correlated with the behavior in multiple other contexts [6]. Research into animal personality has grown over the last decade as its relevance to animal health and welfare has become more apparent [7]. Personality has been used also for aspects of captive management, including decreasing stress [8], increasing positive health outcomes [9], successful breeding, also in terms of infant survival [10,11]. For wildlife management, determining inter-species differences in the personality traits of communally housed animals could be of great help to optimize the use of resources, in order to improve animal welfare [12]. In group-living species, integrated decisions made by individuals result in collective behaviors [13], which may, in turn, influence interactions between individuals and shape the resulting social system [14]. There is evidence that animal groups may exhibit coordinated behavior and make collective decisions based on simple interaction rules [2]. It has been described that in a flock or a colony; birds tend to exhibit behavioral synchrony, maintaining similar behavior at approximately the same time throughout the group [15]. Within-group responsiveness provides important benefits for individual group members, allowing for cohesion and consensus to form in movement decisions [2,16], predator avoidance [17] and resource exploitation [18]. In addition, wild penguins have exhibited within-group synchrony: rockhopper (*Eudyptes chrysocome*) and Adélie (*Pygoscelis adeliae*) penguins synchronized their diving to increase fishing success [19], and cooperative foraging has been reported for African penguins (*Spheniscus demersus*) [20], while little penguins (*Eudyptula minor*) display group behavior arriving at the colony and departing to sea, as a predator avoidance strategy [18]. What about captivity? Some of the behaviors manifested in captive animals may be a consequence of regimented schedules, since captive animals are usually managed at a specific time and place, and fed with similar food items [15]. 

To date, the link between social behaviors expressed by groups and personality remains poorly explored. Recent studies on wild birds have suggested that individual-level personality traits influence the individual degree of use of social information, attraction to conspecifics, and attitude to act as leaders/followers [2]. At the population-level, Kurvers et al. [21] demonstrated that the use of social information in natural populations of barnacle geese decreased with increasing boldness, indicating that personality differences can affect behavioral decisions related to spatial distribution and group formation processes. Michelena et al. [16] investigated the links between animal personality and the individual and collective decision-making processes in groups of sheep. They noticed that shy individuals were found to have a higher social attraction parameter and graze closer to others. 

Does interspecific group life impact on the expression of personality traits on communally housed species? There are no data published on this subject. In this work, we wanted to verify the hypothesis that the penguins of three different species, despite being hosted in common in a dedicated enclosure, express different traits of the personality, and if it is possible to trace, for some traits, a profile that characterizes the species. For this purpose, we have considered three species of penguins, housed together at Royal Zoological Society of Scotland (RZSS) Edinburgh Zoo, Scotland, UK. The exhibit houses a colony of gentoo penguins (*Pygoscelis papua*), a bachelor group of king penguins (*Aptenodytes patagonicus*) and a small colony of northern rockhopper penguins (*Eudyptes moseleyi)*. In the International Union for Conservation of Nature (IUCN) Red List of threatened species, gentoo penguins are classified as least concern, with a wild population estimated number of mature individuals of 774,000 and a population trend referred as stable. Gentoo penguins have a circumpolar breeding distribution that ranges in latitude from the Fish Islands on the Antarctic Peninsula (66°01′ S) to the Crozet Islands (46°00′ S). The three most important locations, containing 80% of the global population, are the Falkland Islands (Malvinas), the South Georgia and the South Sandwich Islands) and the Antarctic Peninsula (including South Shetland Island) [22]. King penguins are also classified as least concern, with a wild population trend referred as increasing; they breed on various sub-Antarctic islands, in the French Southern Territories, Prince Edward Islands (South Africa), Heard Island and McDonald Islands, Macquarie Island (Australia), with small colonies on the Falkland Islands (Islas Malvinas), in southern Chile and in South Sandwich Islands [22]. Regrettably, northern rockhopper penguins are classified as endangered, with a wild population estimates number of mature individuals of 480,600 and a population trend referred as decreasing. Approximately, 85% of the northern rockhopper penguin global population is found in the Atlantic Ocean, breeding at the Saint Helena, Ascension and Tristan da Cunha UK Overseas Territories. The remaining 15% of the population is found in the Indian Ocean on Amsterdam and St Paul islands (French Southern Territories) [22].

According to the Zoological Information Management System (ZIMS), the online database of wild animals maintained in captivity; there are currently 954 recorded gentoos individuals (intended as *Pygoscelis papua* species), housed in 36 zoological institution around the world (17 European, 15 North American, 3 Asian and Auckland Zoo), 532 king penguins individuals recorded, housed in 34 Zoos (15 European, 7 Asian, 11 north American and Auckland Zoo), and only 155 northern rockhoppers, individuals, housed in 10 institutions (6 European, 1 African and 3 north American).

In a mixed species enclosure, animals are far more intermingled than they would be in the wild and have a limited area in which to maintain different territories. Therefore, we wanted to evaluate whether, in these three species of communally housed penguins, it was possible to highlight specie-specific personality traits, despite the influence of forced cohabitation.

The scope of this project was:To provide a preliminary evaluation for the personality traits obtained by the questionnaire filled by the keepers;To investigate if personality traits vary in a sample of three species of communally housed captive penguins;to investigate if it is possible to identify specie-specific personality trends.

## 2. Materials and Methods

### 2.1. Subjects and Housing

The subjects included in the study are briefly described in Appendix A. In total, 43 penguins, randomly selected among the colony housed at RZSS Edinburgh Zoo, Scotland, UK, were considered: 21 northern rockhopper (*Eudyptes Moseley*, 11 m and 10 f); 14 gentoo penguins (*Pygoscelis papua*, 7 m and 7 f) and 9 king penguins (*Aptenodytes patagonicus*, all males). 

The Edinburgh Zoo ‘Penguins Rock’ is the refurbished enclosure opened in 2013. It is 85 m long, 30 meters wide and is equipped with a 65-meter-long pool, which is 3.5 m deep at the deepest point. The pool is freshwater, and holds approximately 1.2 million liters of water, filtered via a filtration system. ‘Penguins Rock’ is one of the largest outdoor penguin exhibits in Europe and it hosts a colony of 129 penguins. There is no foliage incorporated into the enclosure, to eliminate the risk of *aspergillosis* spores affecting the birds. The kings and gentoos are housed together all year round, while northern rockhopper penguins are separated off, in February, and are moved to a smaller ‘creche’ enclosure, attached to the main exhibit, for their breeding season. 

The penguins are fed 3 times a day (first thing in the morning, between 2:30 p.m. and 4 p.m. and between 4:30 p.m. and 5:45 p.m.). The breeding gentoos are also given an extra feed before 1 p.m. The penguins are all hand-fed: this allows keepers to have a close visual health check every day on the birds (look inside beaks, look at toenails, eyes, and feather condition) in order to verify the health conditions and plan timely veterinary medical interventions. If necessary, medical treatments are administered hidden into the gills of the fish and then hand-fed to that penguin. In the colony there are also some old age penguins, which need daily diet supplementation. Enrichments are given regularly according to a scheme.

### 2.2. Personality Assessment 

With the purpose of evaluating personality, we decided to acquire trait-rating assessment from the people who know the animals best (zoo keepers). Following the protocol from Chadwick [8], questionnaires were given to the two keepers who have regular interactions with the penguins, to rate the personality traits of the animals they attend. The test required that the keepers, who both expressed their assessment on all the animals in the study, did not consult during the compilation of the questionnaires. The first part of the questionnaire presented some questions related to keeper’s work, to their interests for the birds (in particular penguins), and to the perception that they have of the personality influence on the animals’ behavior and health. In order to describe penguin behavioral and personality aspects, in the second part of the questionnaire were included 31 adjectives (Appendix A), which were rated on a scale of 1 (trait never exhibited) to 12 (trait always exhibited), depending on how well they described each penguin [23,24]. Two keepers completed the questionnaires for each animal. 

### 2.3. Statistical Analysis 

Concordance between keepers. Inter-rater reliability (IRR) was calculated for each personality trait as Cronbach’s alpha (CA) (SPSS ver. 24 for Windows). IRRs were calculated separately for each species, and then, in the rockhopper and gentoo penguins, repeated separately for females and males. Only traits with an IRR > 0.5 for at least two species were used in subsequent analyses. 

Personality traits. The considered variables were analyzed through descriptive statistics; for each variable, the first, the third quartile, the median value were calculated and then represented by box and whiskers plots (Jamovi for Windows, 0.9.6, Jamovi Computer Software, https://www.jamovi.org/). The box-whisker plots were performed for the three species and the gender.

Differences in traits by species: In order to evaluate differences between species, every species was analyzed for the aforementioned variables; the differences between species were calculated using the Kruskal-Wallis nonparametric test, followed by Steel-Dwass-Critchlow-Fligner test for multiple comparisons (Jamovi for Windows). The statistical significance was set at *p* < 0.05. Species were also compared for each personality trait using a generalized linear model (GLM) in R. Normality of residuals was confirmed for all behavioral traits using the R plot function.

Gender differences: for rockhopper and gentoo penguins a glm was used to investigate the effects of gender, species and their interaction in a 2-way ANOVA design.

Multivariate analysis of personality variables: a multivariate Multiple Factor Analysis (MFA) was applied to the variables (R for Windows, https://cran.r-project.org/bin/windows/) to visualize the differences among species. 

### 2.4. Ethical Statement

All keepers gave their informed consent for inclusion before they participated in the study. The study was conducted in accordance with the Declaration of Helsinki, and the protocol, by its nature, did not require the approval of the Ethics Committee of the Royal Zoological Society of Scotland.

## 3. Results

Two keepers filled out the questionnaires, separately and without consulting each other, for each penguin involved in the study. Both keepers work in contact with penguins for 37.5 h per week, from about 8 years (average value), and enter the enclosure routinely with the penguins. Despite having declared to love animals (penguins in particular), they do not have pets at home.

### 3.1. Inter Rater Coefficient between Keepers 

In the Table 1 and Table 2 are reported the Cronbach’s alpha coefficient (CA), which express the reliability of keepers’ ratings, along with their strength and the statistical significance. These coefficients are calculated in order to highlight the agreement between two or more evaluators, and their values are classified on the basis of their “strength” as follows: <0.5—poor agreement (P); 0.5–0.75—moderate agreement (M); 0.75–0.9—good agreement (G); > 0.9—excellent agreement (E).

### 3.2. Specie Differences in Personality Traits

In Figure 1, the personality traits data graphically resumed by box-whiskers plots is shown; indicating median, 1st and 3rd quartile, non-outlier range and outliers, represented by dots (Jamovi for Windows). The box-whisker plots were performed for all the variables in the three species, but only the personality traits that revealed significant differences to the statistical analysis between the three species considered were included in the figure.

Differences between the median values of the personality traits in the three species were analyzed by the Kruskal-Wallis non-parametric test (Appendix A) followed by the Dwass-Steel-Critchlow-Fligner (DSCF) test for multiple comparisons (Table 3). Only statistically significant results are reported.

The Dwass-Steel-Critchlow-Fligner (DSCF) test for multiple comparisons analysis evidenced that gentoo penguins were significantly more active than the northern rockhopper penguins, which turns out to be significantly calmer (not easily disturbed by changes in the environment) (Figure 1, Table 3). Gentoo penguins were significantly more curious (approaches and explores changes in the environment, enriching and novel objects) than the northern rockhopper (Figure 1, Table 3). Gentoo also scored as less friendly with other penguins (initiates and seems to seek proximity to other penguin) than the kings and northern rockhoppers (Figure 1, Table 3). Gentoo penguins were also rated as significantly more playful (initiates and engages in play behavior, seemingly meaningless, non-aggressive behavior, with objects and/or other penguins) of both other species. In regard to the vocalizations, the gentoos showed a greater propensity to vocalizations than the other species.

### 3.3. Gender Differences in Personality Traits

Analyzing the effect of gender on personality traits, it was possible to highlight a statistically significant difference in the expression between males and females in the traits related to aggression (Figure 2A–C).

Both gentoo and northern rockhopper male penguins were rated as significantly more aggressive to the keepers and to the observer than the females. The trait vocal: aggressive showed a statistically significant difference not only between males and females, but also between the two species analyzed (Figure 2B). In two other traits a gender difference was highlighted, even if the statistical analysis of the values did not reach significance (Figure 3). Both northern rockhopper and gentoo males showed a greater propensity to play than females (Figure 3A). Gentoo female have been rated as more curious than males: the opposite occurred in rockhopper penguins (Figure 3B). For playful and curious traits, the difference between the species was statistically significant.

### 3.4. Multivariate Analysis of Personality Variables

The characteristics of the variables in a multivariate space were analyzed by the multifactor analysis (MFA) technique. The quantitative variables were the personality traits with a Cronbach’s alpha coefficient values classified as moderate, good or excellent in all three species considered (5 traits) or, at list, in two out of three species (2 traits; Table 1); gender and specie were included as qualitative variables.

Figure 4 shows the contribution, expressed as a percentage, of the groups and of the variables considered in the analysis in dimensions 1 (Figure 4A,B) and 2 (Figure 4C,D).

The 3D chart shown in Figure 5 highlights the three dimensions and the 90% nonparametric confidence interval (JMP Ver, 7 for Windows, SAS Institute, Cary, NC, USA). The graph in Figure 5 clearly shows that the personality traits related to aggressiveness and friendliness of the three species are separated in the multidimensional space.

## 4. Discussion

In past years, personality assessment through keeper ratings has been criticized for being too subjective, anthropomorphic and not scientific [25,26], since keepers use their impression and knowledge to judge the animals and their behaviors [27,28]. In the scientific literature, an increasing body of evidence indicates that keeper ratings are both reliable and valid, and there is also little evidence supporting the contention that ratings are tainted by anthropomorphism [29]. Kwan et al. [30] found that the raters were not projecting their characteristics onto their pet dogs, and Weiss et al. [31] found no differences between ratings of chimpanzee personality obtained from American and Japanese observers, indicating that there was no influence of the cultural backgrounds and of the experiences of raters. Observer ratings have been used in assessing the welfare and personality of farm animals [32] and the personality of companion animals [33]. In the same way, keeper ratings can be used to investigate the welfare and personality of zoo animals [23,34]. A fundamental condition in personality studies is that the assessment of personality traits must be both reliable and valid [25,35,36]. Raters scoring the animals must agree in their assessments, and this agreement must be confirmed by testing inter-rater reliability [37]. Therefore, it is fundamental that people who provide ratings for each animal do it independently and do not confer on their answers [25]. According to that, the experimental setting of our study required that the keepers did not consult during the compilation of the questionnaires, this was designed to provide an assessment of the behavioral characteristics of each individual. The analysis of the questionnaires filled out by the keepers showed an intraclass correlation coefficient which allow to further elaborate the data, since a moderate agreement between the two evaluators was highlighted. 

The results obtained in the present study have shown that the keepers knew every penguin very well, and they were able to consistently evaluate their behavioral characteristics [23,24]. This data is useful to underline the importance of the professional figure of the zookeeper, which has prominent responsibilities in the daily welfare of the animals [29,38]. 

In our study, we aimed to analyze the differences in the expression of personality traits of three penguin species housed within the same enclosure, in a controlled environment. The results we obtained indicated that six of the 31 personality traits considered presented statistically significant differences between the species involved. These data point out that the three species of penguins, despite living in close contact, sharing the same enclosure and the same management procedures permanently (or for most of the year), maintain, at least for some personality traits, a species-specific behavioral individuality. Our finding is in accordance with what has been observed by Foerder et al. [15] in an observational study of a colony of 65 penguins of two different species, chinstrap (*Pygoscelis antarctica*) and gentoo (*Pygoscelis papua*) housed in the same environment in the Central Park Zoo (USA). Foerder results indicated that the penguins, despite showing a behavioral synchronism and despite the restrictions imposed by captivity, maintained a behavioral species separation. 

In the last year, space in zoos is becoming more limited by the necessity to increase individual enclosure size: maintaining more than one species in a single enclosure zoo may increase the conservation return on the infrastructure [39]. Moreover, it has been highlighted that mixing species in an enclosure is a mean to increase social complexity, which is a paramount aspect of enrichment for many species [40]. However, this choice also has its downsides: changes in conspecific and species composition can also be a cause of stress. Unlike the domesticated farm animals, zoo animals have not been selected for adaptation to captive condition: conversely, strenuous efforts are implemented specifically to raise animals that behave as similar as possible to their wild living conspecific [40]. The awareness that the cohabitation of the species involved in this study does not induce the flattening of the expressions of the behavioral traits it could be an excellent index of well-being, which would also be interesting to investigate in other respects.

In these colonies, where different species live together, it would be interesting to evaluate the existence of a link between expression of personality traits and the choice of the reproductive partner, the effectiveness of parental care, and the survival of newborns, topics particularly delicate, especially in species considered vulnerable from the conservation point of view.

This approach could be useful in one of the central missions of the modern zoo: to conserve endangered species through captive reproduction and to educate the visitors to conservation issues.

## 5. Conclusions

The results obtained in the present study suggest that, despite close cohabitation and a common managerial routine, that is likely to induce a certain degree of behavioral synchrony [15,18], the penguins maintain their own behavioral individuality. For many of the personality traits we analyzed, it was possible to detect an expression that differs according to the species. From a practical point of view, these data could help the management of the animals, allowing them to design and structure the enclosure according to the different behavioral characteristics of the cohabiting species, and also could improve the relationship strategies with the keepers [41]. Moreover, in everyday life, the awareness of the personality characteristics of individuals would be beneficial in the design of enrichments, which could be tailor-made to the needs of the individual [42]. 

## Figures and Tables

**Figure 1 animals-09-00376-f001:**
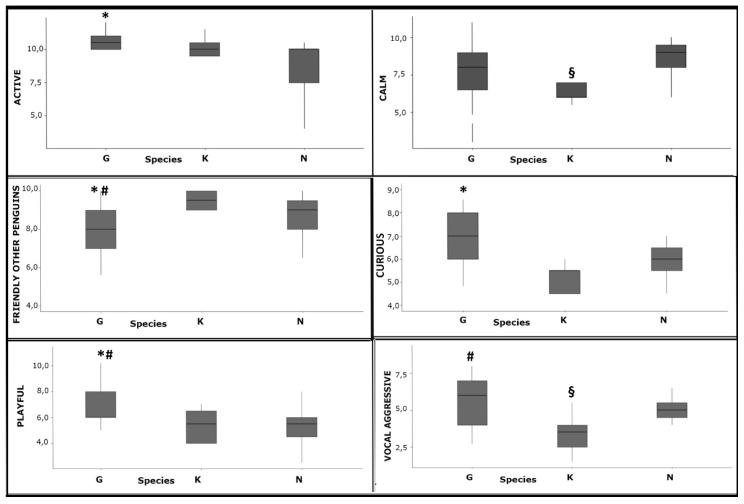
Box-whiskers plots for the statistically different personality traits in the 3 species considered (G = Gentoo penguins, K = King penguins, N = Northern rockhopper penguins), *p* < 0.05 values are considered as statistically different, * = Gentoo vs. northern rockhopper; # = Gentoo vs. king; § = King vs. rockhopper.

**Figure 2 animals-09-00376-f002:**
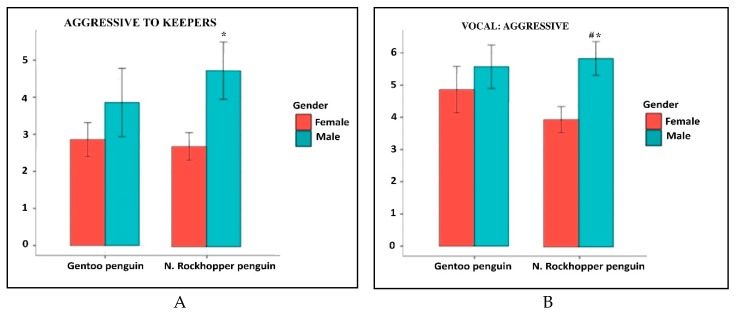
Graphic representation of the traits that presented statistically significant differences between males and females. In (**A**) Aggressive to keepers; in (**B**) Vocal: aggressive; in (**C**) Aggressive to observer. *p* < 0.05 values are considered as statistically different. * = male vs. female; # = gentoo vs. northern rockhopper.

**Figure 3 animals-09-00376-f003:**
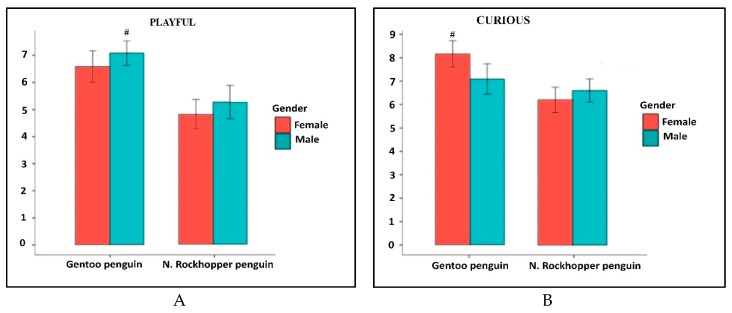
Graphic representation of the traits that showed differences between males and females. with values very close to statistical significance, which was present in the difference between the species. In (**A**) Playful; in (**B**) Curious. *p* < 0.05 values are considered as statistically different. # = Gentoo vs. northern rockhopper.

**Figure 4 animals-09-00376-f004:**
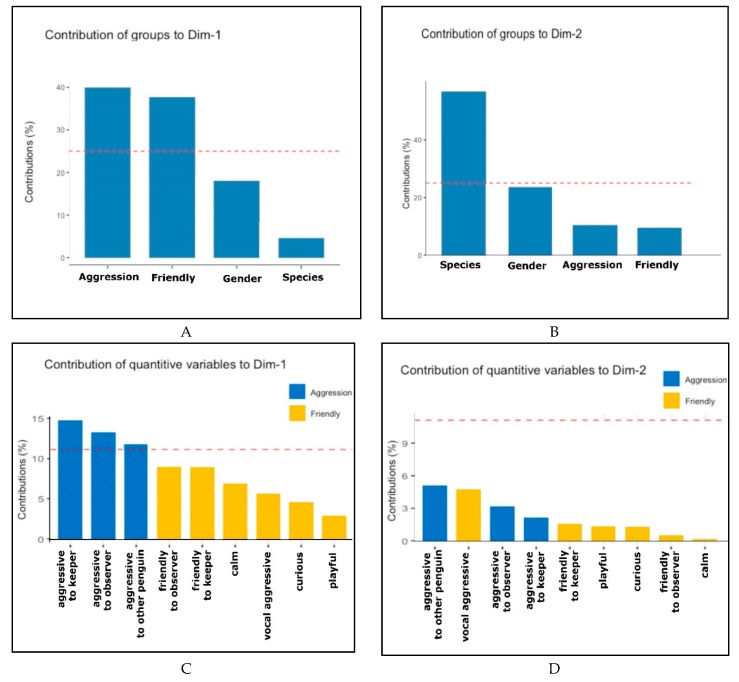
Graphic representation of the contribution (expressed as a percentage) of the groups and of the considered variables in the analysis in Dimensions 1 and 2. In (**A**) Contribution of the groups to Dimension 1; in (**B**) Contribution of quantitative variables to Dimension 1; in (**C**) Contribution of the groups to Dimension 2; in (**D**) Contribution of quantitative variables to Dimension 2. The red line represents the threshold where variables are judged to be contributing ‘significantly’. It is a guideline base on the relative proportions of variance and not a formal significance test.

**Figure 5 animals-09-00376-f005:**
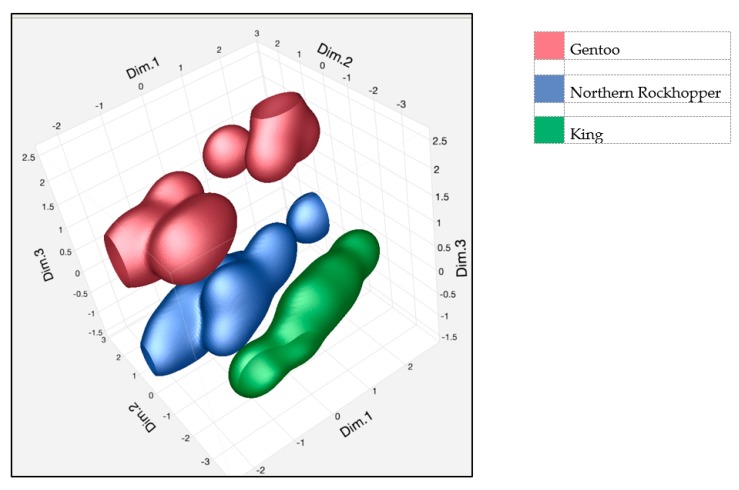
3D graph reporting the three dimensions and the 90% nonparametric confidence interval. It is evident how the three species clusterize and differentiate themselves gravitating around different key personality traits.

**Table 1 animals-09-00376-t001:** The table comprises Cronbach’s alpha coefficient values (CA), calculated for all penguins together (all penguins) and divided by species. The coefficients are classified as “poor” (P), “moderate” (M), “good” (G) and “excellent” (E). *p* < 0.05 values are considered as statistically different. * and ** = trait considered in the subsequent Multivariate analysis of personality variables: * CA value > 0.5 in at least two species of the three considered; ** CA value > 0.5 in all the three species considered.

Trait	All Penguins			Gentoo Penguin			King Penguin			Northern Rockhopper Penguin		
	CA	Strength	*p* Value	CA	Strength	*p* Value	CA	Strength	*p* Value	CA	Strength	*p* Value
Active	0.831	G	0.000	0.225	P	0.326	0.911	E	0.005	0.891	G	0.000
Aggressive to Other Penguin *	0.645	M	0.001	0.906	E	0.000	0.838	G	0.022	0.415	P	0.126
Aggressive to Familiar People	0.589	M	0.003	0.000	P	0.610	0.905	E	0.006	0.625	M	0.013
Aggressive to Keepers **	0.758	G	0.000	0.889	G	0.000	0.977	E	0.000	0.689	M	0.007
Aggressive to Unfamiliar People	0.477	P	0.022	0.000	P	0.585	0.278	P	0.351	0.701	M	0.006
Aggressive to Observer **	0.688	M	0.000	0.917	E	0.000	0.720	M	0.073	0.632	M	0.018
Calm *	0.516	M	0.012	0.509	M	0.106	0.773	G	0.012	0.452	P	0.1
Cooperative	0.479	P	0.021	0.700	M	0.019	0.851	G	0.018	0.018	P	0.485
Curious **	0.630	M	0.001	0.607	M	0.052	0.933	E	0.002	0.788	G	0.001
Dominant	0.142	P	0.316	0.305	P	0.260	0.141	P	0.429	0.117	P	0.394
Eccentric	0.000	P	0.500	0.00	P	0.500	0.00	P	0.5	0.000	P	0.5
Excitable	0.000	P	0.573	0.00	P	0.533	0.00	P	0.729	0.00	P	0.5
Friendly to Other Penguins	0.565	M	0.005	0.744	M	0.01	0.34	P	0.313	0.481	P	0.081
Friendly to Keepers *	0.672	M	0.000	0.815	G	0.002	0.431	P	0.255	0.589	M	0.03
Friendly to Familiar People	0.164	P	0.150	0.000	P	0.5	0.96	E	0.001	0.003	P	0.502
Friendly to Unfamiliar People	0.000	P	0.588	0.000	P	0.5	0.64	M	0.120	0.000	P	0.57
Friendly to Observer **	0.674	M	0.000	0.838	G	0.001	0.675	M	0.099	0.527	M	0.056
Fearful of Other Prenguins	0.000	P	0.603	0.000	P	0.5	0.000	P	0.500	0.000	P	0.625
Fearful of Familiar People	0.015	P	0.481	0.000	P	0.5	0.00	P	0.5	0.398	P	0.139
Fearful of Unfamiliar People	0.000	P	0.677	0.00	P	0.5	0.00	P	0.5	0.00	P	0.722
Fearful of Keepers	0.042	P	0.466	0.00	P	0.5	0.00	P	0.875	0.600	M	0.002
Fearful of You	0.149	P	0.306	0.00	P	0.5	0.000	P	0.790	0.843	G	0.000
Insecure	0.315	P	0.118	0.00	P	0.571	0.585	M	0.154	0.185	P	0.330
Playful **	0.634	M	0.001	0.618	M	0.47	0.548	M	0.178	0.647	M	0.014
Self-Assured	0.246	P	0.188	0.00	P	0.792	0.625	M	0.129	0.259	P	0.260
Smart	0.342	P	0.095	0.225	P	0.327	0.855	G	0.017	0.310	P	0.213
Solitary	0.559	M	0.006	0.037	P	0.474	0.906	E	0.006	0.718	M	0.004
Tense	0.713	M	0.000	0.000	P	0.690	0.993	E	0.000	0.890	G	0.000
Timid/Shy	0.000	P	0.589	0.000	P	0.754	0.676	M	0.098	0.495	P	0.073
Vocal: Aggressive **	0.703	M	0.000	0.772	G	0.014	0.822	G	0.027	0.586	M	0.031
Vocal: Non Aggressive	0.440	P	0.035	0.681	M	0.024	0.908	E	0.005	0.000	P	0.544

**Table 2 animals-09-00376-t002:** The table comprises Cronbach’s alpha coefficient values (CA), calculated separately for Gentoo and Northern Rockhopper, divided by gender. The coefficients are classified as “poor” (P), “moderate” (M), “good” (G) and “excellent” (E), *p* < 0.05 values are considered as statistically different.

Trait	Gentoo Penguin						Northern Rockhopper Penguin					
	Female			Male			Female			Male		
	CA	Strength	*p* Value	CA	Strength	*p* Value	CA	Strength	*p* Value	CA	Strength	*p* Value
Active	0.853	G	0.017	0.000	P	0.785	0.963	E	0.000	0.852	G	0.004
Aggressive to Other Penguin	0.876	G	0.011	0.930	E	0.003	0.203	P	0.370	0.354	P	0.263
Aggressive to Familiar People	0.190	P	0.402	0.000	P	0.563	0.000	P	0.703	0.629	M	0.078
Aggressive to Keepers	0.726	G	0.070	0.920	E	0.004	0.119	P	0.427	0.847	G	0.005
Aggressive to Unfamiliar People	0.286	P	0.347	0.000	P	0.543	0.000	P	0.849	0.699	M	0.044
Aggressive to Observer	0.878	G	0.011	0.936	E	0.002	0.107	P	0.434	0.835	G	0.006
Calm	0.542	M	0.182	0.510	M	0.203	0.560	M	0.119	0.441	P	0.200
Cooperative	0.711	M	0.078	0.671	M	0.101	0.485	P	0.168	0.000	P	0.759
Curious	0.963	E	0.000	0.331	P	0.319	0.853	M	0.004	0.721	M	0.035
Dominant	0.560	M	0.171	0.212	P	0.390	0.114	P	0.430	0.000	P	0.652
Eccentric	0.000	P	0.500	0.000	P	0.500	0.000	P	0.500	0.000	P	0.500
Excitable	0.000	P	0.543	0.000	P	0.500	0.000	P	0.500	0.000	P	0.500
Friendl to Other Penguins	0.857	G	0.016	0.689	M	0.091	0.769	G	0.020	0.042	P	0.475
Friendly to Keepers	0.547	M	0.179	0.902	E	0.006	0.834	G	0.007	0.400	P	0.230
Friendly to Familiar People	0.000	P	0.500	0.000	P	0.500	0.000	P	0.500	0.000	P	0.500
Friendly to Unfamiliar People	0.000	P	0.500	0.000	P	0.500	0.000	P	0.618	0.000	P	0.500
Friendly to Observer	0.606	M	0.141	0.906	E	0.006	0.665	M	0.059	0.450	P	0.193
Fearful of Other Prenguins	0.000	P	0.500	0.000	P	0.500	0.000	P	0.686	0.000	P	0.500
Fearful of Familiar People	0.000	P	0.500	0.000	P	0.500	0.554	M	0.0123	0.000	P	0.500
Fearful of Unfamiliar People	0.000	P	0.500	0.000	P	0.500	0.000	P	0.681	0.021	P	0.500
Fearful of Keepers	0.000	P	0.500	0.000	P	0.500	0.750	G	0.026	0.000	P	0.500
Fearful of Observer	0.000	P	0.500	0.000	P	0.500	0.875	G	0.002	0.000	P	0.500
Insecure	0.000	P	0.725	0.060	P	0.471	0.000	P	0.623	0.000	P	0.500
Playful	0.623	M	0.130	0.574	M	0.162	0.536	M	0.134	0.721	M	0.035
Self-Assured	0.000	P	0.951	0.000	P	0.648	0.000	P	0.692	0.683	M	0.051
Smart	0.765	G	0.051	0.061	P	0.471	0.773	G	0.019	0.000	P	0.717
Solitary	0.075	P	0.463	0.000	P	0.500	0.761	G	0.022	0.652	M	0.066
Tense	0.066	P	0.500	0.910	E	0.001	0.910	E	0.001	0.868	G	0.003
Timid/Shy	0.000	P	0.500	0.000	P	0.702	0.385	P	0.240	0.696	M	0.045
Vocal: Aggressive	0.784	G	0.042	0.625	M	0.129	0.000	P	0.607	0.807	G	0.011
Vocal: Non Aggressive	0.000	P	0.621	0.862	M	0.015	0.000	P	0.697	0.515	M	0.148

**Table 3 animals-09-00376-t003:** Dwass-Steel-Critchlow-Fligner (DSCF), test for multiple comparisons analysis of the differences between the median values of the personality traits, rated by the keepers, in the three species of penguin. W = Wilcoxon rank sum test statistic; *p* < 0.05 values are considered as statistically different. Only statistically different results are listed.

**Pairwise Comparisons—ACTIVE**			
		**W**	***p***
Gentoo penguin	King penguin	−1.15	0.415
Gentoo penguin	Northern rockhopper penguin	−3.98	0.005
King penguin	Northern rockhopper penguin	−1.83	0.195
**Pairwise Comparisons—CALM**			
		**W**	***p***
Gentoo penguin	King penguin	−2.16	0.127
Gentoo penguin	Northern rockhopper penguin	2.49	0.078
King penguin	Northern rockhopper penguin	3.90	0.006
**Pairwise Comparisons—CURIOUS**			
		**W**	***p***
Gentoo penguin	King penguin	−2.255	0.111
Gentoo penguin	Northern rockhopper penguin	−3.251	0.022
King penguin	Northern rockhopper penguin	−0.226	0.873
**Pairwise Comparisons—FRIENDLY TO OTHER PENGUINS**			
		**W**	***p***
Gentoo penguin	King penguin	3.300	0.020
Gentoo penguin	Northern rockhopper penguin	3.109	0.024
King penguin	Northern rockhopper penguin	1.805	0.191
**Pairwise Comparisons—PLAYFUL**			
		**W**	***p***
Gentoo penguin	King penguin	−3.10	0.028
Gentoo penguin	Northern rockhopper penguin	−4.07	0.004
King penguin	Northern rockhopper penguin	−0.32	0.819
**Pairwise Comparisons—VOCAL: AGGRESSIVE**			
		**W**	***p***
Gentoo penguin	King penguin	−3.99	0.005
Gentoo penguin	Northern rockhopper penguin	−1.72	0.225
King penguin	Northern rockhopper penguin	3.48	0.014

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
