# Peer review of "Comparative Personality Traits Assessment of Three Species of Communally Housed Captive Penguins"

_animals, 2019, doi:10.3390/ani9060376_

Round 1

Reviewer 1 Report

Comments to Author

Thank you to the authors for considering the previous feedback. However, there remains a significant issue in the paper that was pointed out in response to the previous draft. Many species and gender comparisons were made using traits that had very poor ICC values. As was previously mentioned, a poor ICC means that the trait was not scored consistently between the raters. If the raters could not apply a term consistently, the validity of that term as a trait describing penguin personality is questionable, and it is not appropriate to make further comparisons regarding that term. After all, if the raters cannot consistently agree how much individual penguins express a different trait, how can that trait be used to make comparisons at the species level? I would recommend simplifying the goals and related analysis in this paper. There are no existing studies on penguin personality, and this paper would have a greater impact if it helped to conclusively establish valid traits that raters can use in future studies of penguin personality. Although the three-dimensional modeling is interesting, it is only meaningful if the underlying data (i.e., the trait ratings) are valid. The authors may also wish to consider how they could provide evidence of the external validity of these ratings, such as by comparing keeper ratings to observational data on behavior.

Specific Comments by Line Number

-          The simple summary seems to overemphasize the topic of behavioral synchrony, which was not the topic of study here. I would keep this simpler and talk about how you are trying to see if there are measureable personality differences in these species. The question of whether expression of personality traits is inhibited in group-living animals is more complicated and is not something the data here actually tested.

-          Line 31: should be “optimizing” not optimize.

-          Line 83: should read “species” not specie. Species is both plural and singular. “Specie” is not a word in English.

-          Line 88: gentoo penguin should not be capitalized, nor should king penguin. Only animal names including proper noun should be capitalized.

-          Line 126: “Northern” is misspelled, and the second word in a species name is not capitalized.

-          Line 168: The effort to correct this aspect of the statistics is appreciated. However, I would still argue that if you want to make a comparison using a particular species, then the ICC has to have a decent value (at least 0.5) for THAT species. Realistically, only traits with a good ICC for all three species should have been used in later analyses for that reason. It is recognized that this does limit the number of terms that can be used, but that is ok. This is a first step towards understanding penguin personality, and it is more important to have a set of results that are valid than to have many analyses.

-          Line 188: This is kind of a strange word choice: “confronting.” I think you mean to say “consulting.”

-          Table 3: The table legend should explain what * and ** refer to. “Friendly” to other penguins is misspelled. “Rockhopper” is also misspelled.

-          Table 4: “Rockhopper” is again misspelled.

-          This table has a large number of very high values listed for Cronbach’s alpha which must represent errors, because the absolute value of Cronbach’s alpha should typically fall between 0 and 1.

-          Line 204: “Species” is misspelled.

-          Lines 227-244: Although the authors indicated that they only did subsequent comparisons using behaviors with an alpha over 0.5 in at least two species (which is still arguably an issue), there many comparisons listed here for traits that showed much worse validity. For example, the authors compare the trait “fearful of familiar people,” among species, which according to Table 4, showed poor interrater reliability for all three species, individually and together. There are several other traits compared here that did not meet the authors’ own criterion for alpha greater than 0.5 for at least two species. Some of these traits are also not mentioned in either Figure 1 or Table 6, despite the fact that these are repeatedly cited.

-          Figure 2, Section 3.3: This section and figure refer to a category “aggressive to observer” repeatedly that is not listed in any of the previous tables. Does this mean “aggressive to keepers” or “aggressive to you”? Depending on which trait this actually corresponds to, there is a good chance the ICC is unacceptably low, so the trait may not have good validity anyway.

-          Figure 4 and 5: Again, behaviors are used regardless of their ICC value here, calling all these results into question.

-          Lines 329-331: This really is not a true statement, because many of the traits used in this analysis had very low ICC values and are likely not particularly meaningful for describing penguin personality.

Author Response

Reviewer 1

We thank the referee for taking the time to help us improve our work. The authors gave the utmost consideration in his/her observations and revised the article according to his/her concerns.

GENERAL COMMENT

Thank you to the authors for considering the previous feedback. However, there remains a significant issue in the paper that was pointed out in response to the previous draft. Many species and gender comparisons were made using traits that had very poor ICC values. As was previously mentioned, a poor ICC means that the trait was not scored consistently between the raters. If the raters could not apply a term consistently, the validity of that term as a trait describing penguin personality is questionable, and it is not appropriate to make further comparisons regarding that term. After all, if the raters cannot consistently agree how much individual penguins express a different trait, how can that trait be used to make comparisons at the species level? I would recommend simplifying the goals and related analysis in this paper. There are no existing studies on penguin personality, and this paper would have a greater impact if it helped to conclusively establish valid traits that raters can use in future studies of penguin personality. Although the three-dimensional modeling is interesting, it is only meaningful if the underlying data (i.e., the trait ratings) are valid. The authors may also wish to consider how they could provide evidence of the external validity of these ratings, such as by comparing keeper ratings to observational data on behavior.

We thank you for your comments. We have further modified the article according to the suggestions received. 

Specific Comments by Line Number

The simple summary seems to overemphasize the topic of behavioral synchrony, which was not the topic of study here. I would keep this simpler and talk about how you are trying to see if there are measureable personality differences in these species. The question of whether expression of personality traits is inhibited in group-living animals is more complicated and is not something the data here actually tested.

 The simple summary was amended in accordance with the reviewer's suggestions.

Line 31: should be “optimizing” not optimize.

The error was corrected.

Line 83: should read “species” not specie. Species is both plural and singular. “Specie” is not a word in English.The error was corrected.

Line 88: gentoo penguin should not be capitalized, nor should king penguin. Only animal names including proper noun should be capitalized.

The error was corrected throughout the manuscript. 

Line 168: The effort to correct this aspect of the statistics is appreciated. However, I would still argue that if you want to make a comparison using a particular species, then the ICC has to have a decent value (at least 0.5) for THAT species. Realistically, only traits with a good ICC for all three species should have been used in later analyses for that reason. It is recognized that this does limit the number of terms that can be used, but that is ok. This is a first step towards understanding penguin personality, and it is more important to have a set of results that are valid than to have many analyses.                                                                          

We have corrected the results, eliminating some graphs. 

Line 188: This is kind of a strange word choice: “confronting.” I think you mean to say “consulting.”                       The sentence has been modified by replacing the word indicated. 

Table 3: The table legend should explain what * and ** refer to. “Friendly” to other penguins is misspelled. “Rockhopper” is also misspelled.

Spelling errors have been corrected and the requested information has been added.

Table 4: “Rockhopper” is again misspelled.

The spelling error has been corrected.

This table has a large number of very high values listed for Cronbach’s alpha which must represent errors, because the absolute value of Cronbach’s alpha should typically fall between 0 and 1.

We apologize. We made some transcription errors, which we have corrected.  

The actual values were:

8.882 was 8.882 E-16; 

1.776 was 1.775 E-12; 

3.775 was 3.775 E-12; 

1.332 was 1.332 E-15;

6.66  was 6.66 E-15;

8.438 was 8.438 E-15.

Line 204: “Species” is misspelled.

The spelling error has been corrected.

Lines 227-244: Although the authors indicated that they only did subsequent comparisons using behaviors with an alpha over 0.5 in at least two species (which is still arguably an issue), there many comparisons listed here for traits that showed much worse validity. For example, the authors compare the trait “fearful of familiar people,” among species, which according to Table 4, showed poor interrater reliability for all three species, individually and together. There are several other traits compared here that did not meet the authors’ own criterion for alpha greater than 0.5 for at least two species. Some of these traits are also not mentioned in either Figure 1 or Table 6, despite the fact that these are repeatedly cited.                                     The paragraph has been corrected as requested.

Figure 2, Section 3.3: This section and figure refer to a category “aggressive to observer” repeatedly that is not listed in any of the previous tables. Does this mean “aggressive to keepers” or “aggressive to you”? Depending on which trait this actually corresponds to, there is a good chance the ICC is unacceptably low, so the trait may not have good validity anyway. Following the reviewer's suggestions, we amended the table, making the trait denomination consistent throughout the manuscript. 

Figure 4 and 5: Again, behaviors are used regardless of their ICC value here, calling all these results into question. We have corrected the results, eliminating some graphs. We have maintained figure 4 and the 3D graph of species differences because they derived from traits with ICC values higher than 0,5 in all three species (5 traits) or in two of the three species (3 traits). Moreover, although aware of the limitations of the analysis, evaluating the ICC of these three traits of all the animals not separated among the species, they had values superior to 0,5.  

Lines 329-331: This really is not a true statement, because many of the traits used in this analysis had very low ICC values and are likely not particularly meaningful for describing penguin personality.

The statement has been amended.

Reviewer 2 Report

Accept in present form

Author Response

We thank the referee for taking the time to help us improve our work. 

Round 2

Reviewer 1 Report

Based on the requested changes made to this manuscript, I am in support of accepting it for publication. 

This manuscript is a resubmission of an earlier submission. The following is a list of the peer review reports and author responses from that submission.

Round 1

Reviewer 1 Report

In this manuscript, the authors assessed if it’s possible to identify specie-specific personality traits, in a colony of three species of penguins hosted in the same enclosure, despite the forced cohabitation. They evaluated the penguins’ personality by given questionnaires filled by zookeepers who had regular interactions with the animals. The paper is clearly motivated and written, and the insight reported is a useful contribution for the research into animal personality, and consequently for health and welfare of zoo animals.

Please see below for a few comments and suggestions that the authors could consider for revisions:

Introduction

(lines 96-102) The authors could improve this part on the conservation status of the three species, with information about environmental and climatic condition of the species. Do the three species live in the same type of natural habitat with the same climatic conditions? I think it is important to specify that in the artificial environment of the enclosure of the zoo the needs of all the three species are met.

Material and methods

Subjects and housing

Table 1: I suggest to move the Table 1 to Supplementary Material section.

Statistical analysis

Please specify why a non-parametric test (Kruskal-Wallis) to assess the differences in traits by species was used. Besides, please specify if the data were tested for normality and equality of variance and which tests were used.

In order to evaluate gender differences, I suggest to use a different statistical test to compare two groups of data, e.g., t-test or Mann-Whitney test.

I suggest to add a table in Supplementary Material section, with the list of the 31 personal traits and the extended explanation of trait (e.g. calmer: not easily disturbed by changes in the environment).

Results

The results of DSCF test for multiple comparisons could be indicated on the figure 1, the authors can insert the statistical significance (i.e. t test P value < 0.05) annotations on top of box plots, and the table 4 could be move in Supplementary Material section.

Please specify in the captions of table 3 and table 4 the meaning of each column.

Did the authors find specific gender differences of traits for Northern rockhopper or Gentoo penguins?

Reviewer 2 Report

Dear Authors, 

The submitted manuscript represents an emerging area of research (nonhuman animal personality) studied in new species that in my experience, have a lot of personality. The authors appropriately identify that the topic has great potential applications for improving husbandry practices that impact the welfare of captive penguins. However, I have some major concerns about the statistical analysis employed that must be addressed to ensure that the proper conclusions are reached. The discussion could also use extensive revision, as it reads more like a summary of results rather than putting the results in the context of the extant literature on this topic. Some specific comments follow by section. 

Introduction

- The dual goals of describing personality dimensions within each species and then comparing species-level characteristics are a little confusing as presented in the introduction and could use some clarification. 

- The last few paragraphs should be revised into full paragraphs - sometimes only a single sentence is considered a paragraph. 

- Given that you did not actually measure behavioral synchrony among the species represented in this study, I am not sure this should be listed as such a goal of the research and emphasized as much in the introduction.

Materials and Methods

- What was the rearing history of these individuals? If the chicks were hand-reared by keepers, this could likely affect many of the variables measured as "personality" traits, particularly those related to social interactions with humans and other penguins. Differences between rearing history among individuals should be taken into account in analysis or at least reported. 

- Did the same two zookeepers complete surveys for all the penguins, or did two keepers rate each penguin, but not necessarily the same two keepers for each penguin? Please clarify the exact number of raters. If it varied between penguins, this should be taken into account when calculating the intraclass correlation coefficients (ICC).

- It seems to me that the ICC analysis should have been conducted separately for each species. Which traits represent valid measures of personality likely varies between species, and lumping all the species together for this analysis likely obscured some of the species-level personality tendencies that were the stated focus of study. 

- Another major statistical issue is that all of the 31 traits seem to have been used for the factor analysis. The purpose of conducting the ICC first is to identify traits that seem to have validity based on the fact that multiple raters score them similarly. Using all the traits in the factor analysis, including those with poor ICC values, means the factor analysis may be based on traits that are not meaningful, calling these results into question. 

- More information in general should be provided in the methods as to how the species were or were not grouped at different levels of analysis.

- I would strongly argue that combining all species to examine gender differences is also a major flaw of the analysis. This would be questionable even if both sexes were equally represented in the study, because again, the study purports to find evidence for species level differences in personality tendencies. Furthermore, the fact that only male king penguins are included in this analysis means that the results are even further skewed when lumping all the species together. I would recommend reanalyzing gender differences for gentoo and rockhopper penguins separately. 

Results

- Table 1 should include ICC values for all 31 traits, and only those traits that showed significant ICC values should be used in further comparisons, such as the information presented in Figure 1 and Table 3. 

- Table 4 contains a number of errors; for example, the trait "active' is listed in three separate places with different values attached, and some of the other trait headers seem to be incorrect as well. 

- I question the utility of "friendly to you" as a  useful trait in addition to friendliness towards keepers and other people.  

- I assume Table 6 refers to the gender differences examined in Figure 2, but the table legend does not explain this fact and, as previously mentioned, this analysis should be done only for the two species that had both sexes represented, and should be separate for those species as well (only analyze one species at a time). 

- The colors chosen to represent the different species are very hard to differentiate in Figure 3 when the paper is printed in greyscale.

Discussion

- The discussion could use some revision. There is no discussion at all of the factor analysis, and instead it reads almost more like a list of results comparing individual traits, which is already in the results section. More effort is needed to place the results of this study in the context of the current literature on nonhuman animal personality.